# The Regulation of Cyclins and Cyclin-Dependent Kinases in the Development of Gastric Cancer

**DOI:** 10.3390/ijms24032848

**Published:** 2023-02-02

**Authors:** Aadil Javed, Mahdieh Yarmohammadi, Kemal Sami Korkmaz, Teresa Rubio-Tomás

**Affiliations:** 1Department of Bioengineering, Faculty of Engineering, Cancer Biology Laboratory, Ege University, Izmir 35040, Turkey; 2Department of Biology, Faculty of Sciences, Central Tehran Branch, Islamic Azad University, Tehran 33817-74895, Iran; 3School of Medicine, University of Crete, 70013 Herakleion, Crete, Greece

**Keywords:** gastric cancer, Cyclins, Cyclin-dependent kinases (CDKs), cell signaling, molecular oncology, kinases, cell cycle regulation

## Abstract

Gastric cancer predominantly occurs in adenocarcinoma form and is characterized by uncontrolled growth and metastases of gastric epithelial cells. The growth of gastric cells is regulated by the action of several major cell cycle regulators including Cyclins and Cyclin-dependent kinases (CDKs), which act sequentially to modulate the life cycle of a living cell. It has been reported that inadequate or over-activity of these molecules leads to disturbances in cell cycle dynamics, which consequently results in gastric cancer development. Manny studies have reported the key roles of Cyclins and CDKs in the development and progression of the disease in either in vitro cell culture studies or in vivo models. We aimed to compile the evidence of molecules acting as regulators of both Cyclins and CDKs, i.e., upstream regulators either activating or inhibiting Cyclins and CDKs. The review entails an introduction to gastric cancer, along with an overview of the involvement of cell cycle regulation and focused on the regulation of various Cyclins and CDKs in gastric cancer. It can act as an extensive resource for developing new hypotheses for future studies.

## 1. Introduction to Gastric Cancer

### 1.1. General Epidemiology of Gastric Cancer

Among malignancies, gastric cancer is considered the fifth most commonly occurring disease, with a million new annual cases. As the late diagnosis of the disease is common, the mortality rate is higher, and for this reason, gastric cancer is the third deadliest cancer worldwide [1]. South America, Eastern Europe, and East Asia are considered hotspots for mortality and incidence of gastric cancer [2]. Cases occur twice as often in men as in women. The recent trend highlights a decreasing mortality rate. However, due to aging populations, the incidence rate of the disease is expected to increase [3].

### 1.2. General Risk Factors for Gastric Cancer

Infection rates for the major cause of the disease, *H. pylori*, have decreased due to economic development and improved living standards [4,5]. Non-cardia gastric cancer is associated with *H. pylori* infection, where the gastric mucosa is subjected to chronic infection leading to atrophic gastritis, which further results in intestinal metaplasia [6]. Most *H. pylori* infections remain asymptomatic. However, gastric cancer occurrence due to infection is associated with the virulence, host, and environmental factors along with oncogene activation causing pathogenicity due to Cytotoxin-Associated Gene A (Cag-A) protein capable of affecting a cohort of cellular signaling pathways [7,8,9]. Treatment of the infection can help reduce the burden of gastric cancer transformation; however, it is dependent on the degree of pre-existing damage [10]. High-risk population environments, familial predisposition, alcohol consumption, cigarette smoking, and older age are other risk factors for non-cardia gastric cancer beyond *H. pylori* infection [11]. *H. pylori* infection has been associated with salted food intake and gastric cancer development; therefore, dietary modification involving food intake with less salt is one of the preventive measures against gastric cancer. High intake of vegetables and fruits is also recommended [12].

### 1.3. Genetic Parameters Associated with Gastric Cancer

The genetic background as an etiological feature is also dominant in gastric cancer, as 1–3% of the cases possess germline mutations and approximately 10% of the patients present familial association [13,14,15]. The three major types of gastric cancer involving autosomal dominance as the hereditary element include gastric adenocarcinoma with proximal polyposis (stomach), familial intestinal gastric cancer, and hereditary diffuse type gastric cancer (HDGC). HDGC cases show a representation of 30–40% of the patients with *CDH1* (E-Cadherin) mutations with screening for *CDH1* mutations now available at expert centers with clinical geneticists. The pathogenic *CDH1* mutation-carrying patients are recommended prophylactic total gastrectomy [16,17,18]. A germline mutation in *CTNNA1* (encoding alpha-E-catenin) is associated with three families of HDGC [19,20]. HDGC also exhibits mutations in *DOT1-L*, *FBXO24*, *IR* and *PALB2* [21,22]. The proximal polyposis carrying gastric adenocarcinoma is linked with APC promoter 1 gene mutations [23]. There are genetic disorders that have been associated with the development of gastric cancer and include PeutzJegher syndrome, (STK11), *MUTYH*-associated adenomatous polyposis, Li-Fraumeni syndrome (TP53), juvenile polyposis (*SMAD4*, *BMPR1A*), Cowden Syndrome (*PTEN*), Lynch syndrome (*PMS2*, *MLH1/2*, *MSH6*), and familial adenomatous polyposis (*APC)* [24].

### 1.4. General Classification of Gastric Cancer

Borrmann classified early gastric cancers according to microscopic appearance, comprising four types, namely: Type I (broad base and polyploid without ulceration), Type II (sharp margins and elevated borders with ulceration), Type III (infiltration at the base with ulceration), and Type IV (thickened wall with diffusive infiltration) [25]. The therapy response along with prognosis according to histopathological phenotypes is a subject of conflict among researchers as variations in histopathological classifications remain [26]. For example, the World Health Organization (WHO) classification leads to complexity due to the distribution into multiple subtypes even for rare cases. The WHO system includes Paneth cell type, hepatoid, medullary carcinoma with lymphoid stroma, signet ring cell, poorly cohesive, mucinous, mucoepidermoid, micropapillary, papillary, parietal cells, and tubular subtypes [27]. Another classification system employs the Lauren system, consisting of distinguishing unclassifiable, indeterminate, diffuse type, and intestinal type gastric cancer. However, many subtypes remain outside the Lauren classification [28]. Furthermore, the Nakamura system of classification is based on undifferentiated and differentiated types, which can inform the decision for endoscopic resection in the cases of early forms of gastric cancer [29].

### 1.5. General Diagnosis and Therapies for Gastric Cancer

The diagnosis of gastric cancer is based on digital image analysis and liquid biopsies [30], with symptoms at the time of presentation including abdominal pain, weight loss, early satiety, anorexia, and indigestion. Late-stage symptoms often include anemia or dysphagia [31]. Surgery remains the foremost defense against gastric cancer and involves endoscopic resection and gastrectomy along with lymphadenectomy and minimally invasive surgery [32]. Other curative options available to patients of gastric cancer are neoadjuvant and perioperative chemotherapy along with preoperative and postoperative radiation therapy, and postoperative adjuvant chemotherapy [33]. In advanced cases of gastric cancer, chemotherapy leads to improvement in quality of life along with survival, especially in cases of metastatic or unresectable forms of gastric cancer [34]. Supportive care provides a 3–4-month survival advantage, whereas combination chemotherapy in patients has been shown to give an approximately 1-year median overall survival [35,36]. Chemotherapy is recommended for patients exhibiting organ function and adequate performance function [37]. The chemotherapeutics routinely used against gastric cancer include Oteracil (orotate phosphoribosyltransferase), Gimeracil (dihydropyrimidine dehydrogenase inhibitor), and S-1 (oral 5-fluorouracil prodrug) along with Irinotecan, Taxanes, and platinum. These therapies can be employed in combination as well [38,39].

## 2. Overview of Cell Cycle Regulation

The cell cycle of a eukaryotic cell has an interphase containing three phases, the G1, S, and G2 phases, and the mitosis (M) phase carrying cytokinesis [40]. During the interphase of the cell cycle, the growth of cells occurs due to the accumulation of the nutrients required for subsequent mitosis, along with the replication of cellular contents including organelles and DNA. Mitosis then occurs, dividing the cells into two daughter cells carrying genetic material and organelles [41]. There are several cell cycle checkpoints ensuring the correct progression of the cell cycle in terms of cellular components and division [42]. Cyclin-dependent kinases (CDKs) are activated by different Cyclins due to their sequential expression throughout the cell cycle [43].

Mitogenic signaling gives rise to the activation of Cyclin D in the G1 phase of the cell cycle, which in turn functions in activating CDK4/6, causing the phosphorylation of retinoblastoma protein (RB) [44]. The E2F transcription factors are then released to activate Cyclin E and Cyclin A transcriptions, leading to further phosphorylation of RB. There are internal CDK inhibitors in cells with families including INK4 and CIP/KIP, which bind the complexes of Cyclin-CDKs to inhibit phosphorylation of RB and cause eventual prevention of E2F release. These mechanisms cause the replication machinery to be arrested, hence acting as a checkpoint for the cell cycle control [45,46]. The mitotic checkpoint is triggered due to various interactions of Cyclin B-CDK1 complexes, which inhibit the RB by phosphorylations from the S to G2/M transition. The Cyclin B/CDK1 complex assists in nuclear envelope breakdown, chromosome condensation, and mitotic spindle formations. The phosphorylations of tumor suppressor RB control the checkpoints, because hyperphosphorylations due to altered signaling in tumor cells render the growth of cells independent of outside or mitogenic signals [47]. A general overview of the cell cycle pathway involving various Cyclins, CDKs, E2F and Rb states is given in Figure 1. In diseased states such as cancers, these pathways are dysregulated, and this knowledge can be utilized to perform diagnosis and targeted therapy [48].

## 3. The Role of CDKs in Cell Cycle Control

The cells in adult tissues carrying diploid DNA usually reside in a state of quiescence marked by the G0 phase that is either permanent or temporary depending on the functionality of the cells. When cells enter a new cell cycle after mitosis, growth factors or hormones trigger a cascade of signaling events that converge on CDK4 or CDK6 to cause the cells to go through another round of cell cycle upon entry into the S phase. The Cyclin D protein with fluctuating levels throughout the cell cycle binds the CDK4/6 to form complexes [49,50]. Activated Cyclin D-CDK4/6 complexes phosphorylate RB along with p130 and p107, which in hypophosphorylated forms function in the recruitment of repressors of E2F transcription factors to inhibit the G1 to S transition. Cyclin C/CDK3 complexes are also involved in RB phosphorylations, which can lead to entry into the S phase directly from Go, which is either a reversible or permanent phase depending on the cell type [51,52]. The downstream target genes of E2F activated upon RB phosphorylated by CDK4/6 include *CCNE*, *CCNA* and *CCNB*, encoding Cyclin E, Cyclin A, and Cyclin B, respectively. Other proteins activated upon this cascade include dihydrofolate reductase (DHFR), ribonucleotide reductase M1 (RRM1), RRM2, and Polo-like kinase 1 (PLK1), spindle checkpoint protein MAD2, and mitotic checkpoint serine-threonine kinase (BUB1). These proteins function in the progression of the cell cycle [53,54].

In late G1, E2F factors activate Cyclin E1 and Cyclin E2, which bind CDK2 to activate the Cyclin E/CDK2 complex, activating CDK2, which before Cyclin E binding is kept sequestered in p27 and p21 bound forms. These inhibitors are also marked for ubiquitin-mediated degradation upon activation of CDK2. CDK2 is further activated by cell division cycle 25A (CDC25A), which dephosphorylates CDK2 [55]. CDK2 phosphorylates a range of substrates to progress the cell cycle and carry out different functions such as histone synthesis (it is a nuclear protein activator of histone transcription), centrosome duplication, and DNA replication [56]. The schematic regulation of the cell cycle by CDKs has recently been reviewed in detail [57]. The activation of CDK4/6 and CDK2 and their downstream targets render the fate of the cell cycle independent of the mitogenic signals; therefore, the restriction point is passed at this point. When the S phase nears its end, Cyclin A displaces Cyclin E from CDK2 and F-box/WD repeats carrying protein 7 (FBXW7), leading to proteasomal degradation of Cyclin E [58].

The termination of the S phase is led by E2F1, and CDC6 phosphorylations by CDK2 bound with Cyclin A, causing entry into the G2 phase of the cell cycle. CDK2 is also involved in the activation of CDK1, which drives the G2/M transition. Cyclin B activates CDK1 and CDK1 phosphorylations, causing nuclear envelope breakdown, chromosome condensation, and mitotic spindle assembly during mitosis. Anaphase-promoting complex/cyclosome (APC/C) degrades Cyclin B and decreases CDK1 activity in the anaphase at the spindle assembly checkpoint (SAC) [59,60]. A high fidelity of genomic integrity is ensured through the realization of the critical events that are dependent on cell cycle progression and mitotic onset, which is enabled by CDK1 rendering it the most important of the Cyclin-dependent kinases [61]. CDK1 activity is further controlled by Myelin transcription factor 1 (MYT1), G2 checkpoint kinase (WEE1), and CDC25C phosphatase. CDK1 is inhibited by phosphorylations at Tyr15 by Wee1, and Thr14 and Tyr15 by MYT1, and these phosphates are removed by CDC25C for activation of CDK1 [62,63].

DNA damage during the cell cycle causes cell arrest to repair the damage before entering mitosis. These checkpoints occur before and after DNA synthesis in G1 and G2 phases. DNA damage signaling involves checkpoint kinases CHK1 and CHK2, along with Ataxia telangiectasia mutated kinase (ATM), Ataxia telangiectasia, and Rad3 related kinase (ATR), and phosphatidylinositol 3-kinase (PI3K) [64]. CHK1/CHK2 axis is activated upon phosphorylation of ATM/ATR kinases, which detect DNA damage [65]. CHK2, upon activation by upstream kinases, leads to p53 activation, causing the G1 arrest to repair the damaged DNA [66]. Cyclin E-CDK2 complex activity is reduced upon activation of p21 by p53, as DNA repair machinery is upregulated upon p21 activation. Eventually, apoptosis is triggered if the stress of DNA damage is not repaired properly [67,68]. CHK1 also triggers G2 arrest by phosphorylating CDC25A to induce proteolysis by ubiquitination. Another substrate of CHK1 is WEE1, which is another mechanism for G2 arrest of the cell cycle [69,70].

The mitotic checkpoint (also known as the spindle assembly checkpoint (SAC)), functions to monitor the metaphase plate in order to correct the chromosome alignment. The checkpoint is regulated by monopolar spindle 1 (MPSI), also known as TTK protein kinase, which regulates the recruitment of checkpoint proteins to the kinetochore complexes. The genome integrity and appropriate chromosome segregation are regulated by the phosphorylation of substrates of TTK [71,72]. The passage of SAC brings the APC/C complex into play, which stimulates the degradation of securing and Cyclin B to initiate mitosis [73,74]. The checkpoints present at various time points throughout the cell cycle ensure the genomic and mitotic fidelity of the cells entering the new cell cycle, and the convergence of various signaling pathways and these checkpoints on CDKs indicate the importance of Cyclins and CDKs in the cell cycle progression. The cell cycle checkpoints and the major regulators of CDKs are depicted in Figure 2. Since dysregulated cell cycle regulation is considered one of the hallmarks of cancer, CDK inhibition for therapeutic purposes, especially CDK4/6 using small molecule inhibitors, presents a convincing solution for overcoming resistance to conventional drugs [75].

## 4. The Regulation of Cyclins and CDKs in Gastric Cancer

As mentioned previously, cell cycle regulators such as Cyclins and CDKs can be utilized as biomarkers, prognostic markers, and diagnostic factors, and targeted using small molecule inhibitors [76,77]. However, there is a need to develop a thorough understanding of not only the downstream effects of inactivation of these regulators, but also the upstream regulators functioning to either activate or inhibit Cyclins and CDKs in specific contexts such as solid tumors or hematological malignancies [78]. Therefore, we constructed this review, in which we describe all the regulators of Cyclins and CDKs studied in gastric cancer cells, tissues or models.

### 4.1. The Regulation of Cyclin D1 in Gastric Cancer

Cyclin D and its isoforms are the first major type of Cyclin taking part in sequentially organizing the order of the events of cell cycle regulation [79]. It activates CDK4/6 upon transcription signals activated via mitogenic or growth factors and is dysregulated in various cancers [80]. Cyclin D1 expression at high levels has been associated significantly with poor prognosis of gastric cancer, as observed in gastric cancer samples in which CD44 expression was also higher, indicating a coexpression of these proteins could be a potential biomarker for the severity of the disease [81].

#### 4.1.1. Factors Downregulating Cyclin D1 in Gastric Cancer

In the natural paradigm of cellular signaling, the events converge at major regulators to keep the cell fates in check. Various factors downregulate the activity of Cyclin D1 in gastric cancer cells and are discussed here in detail. The carbohydrate response element binding protein (CHREBP) is downregulated in gastric cancer and inhibits Cyclin D1 levels in gastric cancer cells in order to suppress growth via negative modulation of the Cyclin D1-Rb-E2F1 signaling mechanism [82]. A ring finger domain-containing protein (RN181), which functions as an E3 ubiquitin ligase, was shown to be downregulated in gastric cancer as compared to normal tissues and regulates Cyclin D1-CDK4 activity via inhibition of ERK/MAPK signaling and G1 to S phase transition [83]. The miR-129-5p induces cell cycle arrest in vitro and in vivo conditions in gastric cancer and targets HOXC10 directly for its downregulation, which further functions in regulating the expression of Cyclin D1 [84]. In another study, involving gastric cancer cell line SGC-7901, curcumin treatment led to the stabilization of miR-34a, which in turn inhibited Cyclin D1 and CDK4 from inducing cell cycle arrest and apoptosis [85]. miR-623 is downregulated in gastric cancer and can sensitize invasive cancer cells to 5-FU for induction of apoptosis. miR-623 directly targets Cyclin D1 mRNA for its down regulation, and *CCND1* overexpression reverses the effects of miR-623 [86].

An Interleukin 6 cytokine family member known as Leukemia inhibitory factor (LIF) downregulates Cyclin D1 and upregulates p21 in gastric cancer cells under both in vivo and in vitro conditions [87]. Tripartite motif-containing 58 protein (TRIM58) is a potential tumor suppressor protein in gastric cancer, where it is presented in the downregulated form. Its overexpression in gastric cancer cell lines led to a reduction of survivin, Cyclin D1, c-myc, and β-catenin. It was concluded that TRIM58 increases β-catenin degradation to inhibit the progression of gastric cancer through ubiquitination-mediated mechanisms [88]. A potential antitumor compound, Glycyrrhizic acid, has been tested for its efficacy in gastric cancer cell lines, and led to G1 arrest and induction of apoptosis, along with reduction of Cyclin D1, D2, D3, E1, and E2 and an increase in pro-apoptotic cleavage patterns of pro-caspases [89]. Dihydroartemisinin (DHA) exerts its anti-tumor activities through inhibition of CDK4 activity and targets Cyclin D1 negatively for induction of cell cycle arrest under in vivo and in vitro conditions in gastric cancer [90]. Table 1 illustrates and summarizes the upstream molecules or inhibitors that act in negative regulation of Cyclin D1 in gastric cancer cells or tissues.

#### 4.1.2. Factors Upregulating Cyclin D1 in Gastric Cancer

There are various upstream regulators of Cyclin D1 that act as oncogenes for the progression of normal gastric cells to gastric carcinoma. Cytotoxin-associated gene A (*CagA*) has been recognized as one of the factors associated with gastric cancer and Cag+ cells exhibit higher expression of Regeneration gene 3 (*Reg3*). Both genes function in the regulation of the cell cycle via Cyclin D1-CDK4 complex formation to induce proliferation and G1 to S transition in gastric cells [93]. Kruppel-like factor 5 (KLF5) is correlated with a worse prognosis of gastric cancer and is positively associated with Cyclin D1 in gastric cancer, as observed in MGC803 and SGC7901 cell lines [94]. Aurora B functions to promote cytokinesis and cell division and has been implicated as one of the promoters of gastric cancer. It facilitates the phosphorylation of H3 at S10, which in turn leads to the transcription of CCND1 and upregulation of Cyclin D1 under in vitro and in vivo conditions in gastric cancer. Moreover, AZD1152, which specifically inhibits Aurora B, also inhibits Cyclin D1 to inhibit tumorigenesis and helps target gastric cancer [91].

An m6A methyltransferase (METTL16) that modifies RNA via N6-methyladenosine (m6A) functions to upregulate Cyclin D1 transcription through its methyltransferase activity in gastric cancer cell proliferation [95]. Epstein–Barr virus has been associated with gastric carcinomas and EBV oncogene BARF1 stabilizes and interacts with Cyclin D1 at transcriptional and protein levels in gastric cancer [96]. Cyclin-dependent kinase 5-regulatory subunit-associated protein 3 (CDK5-RAP3), also known as C53, forming different isoforms, has also been implicated in various cancers, and its isoform d (IC53d) is found upregulated in gastric cancer. IC53d promotes gastric cancer cell proliferation and invasive phenotypes via the promotion of GSK3b and Akt phosphorylations, further increasing Cyclin D1 and G1 to S phase transition [97]. Another oncogene giving rise to WDR5 protein is increased in gastric cancer, where it induces H3K4me3 and Cyclin D1 for the progression of the cell cycle and tumorigenesis [98]. Cyclin D1 protects gastric cancer cells from Doxorubicin, where p73 modulates Cyclin D1 levels positively with the help of activator protein 1 (AP-1) [99].

Another oncogene named *Sine oculis* homeobox homolog 1 (SIX1) is involved in positively regulating the expression of Cyclin D1 along with epithelial–mesenchymal transition (EMT)-related proteins and p-ERK and MMP2 in gastric cancer cells. SIX1 regulation of Cyclin D1 implies that Cyclin D1 is at the crossroads of multiple cellular signaling pathways involved in the progression of gastric cancer [100]. In MGC-803 cells, Resveratrol treatment leads to the inhibition of Cyclin D1, along with c-myc and β-catenin, indicating that Cyclin D1 regulation is influenced by c-myc and β-catenin in gastric cancer cells [101]. Methyltransferase-like 3 (METTL3) modifying mRNA via N6-methyladenosine is involved in gastric cancer, whereby it upregulates Cyclin D1 levels and activates the Akt signaling pathway, as shown in loss-of-function experiments in gastric cancer cells [102].

LINC0857 is a long non-coding RNA that upregulates Cyclin D1 and Cyclin E1 for the progression of gastric cancer and has been indicated as a potential biomarker for the prognosis and diagnosis of the disease [103]. A circular endogenous RNA, has-circ00000647, interacts with and inhibits miR-326-3p and upregulates Cyclin D1 in gastric cancer cells (SGC-7901), showcasing another layer of gene expression regulation upstream of Cyclin D1 for the progression of gastric cancer [104]. Cheng et al. reported that a coronin-like actin-binding protein 1C (CORO1C) functioning in the assembly of F-actin via actin-dependent processes promotes Cyclin D1 and Vimentin for the induction of tumorigenesis in gastric cancer cells [105]. A long non-coding RNA (HOTAIR) targets miR-454-3p to upregulate the STAT3/Cyclin D1 in gastric cancer cells to promote the proliferation and progression of cancer [106]. Table 2 summarizes the molecules involved in the positive regulation of Cyclin D1 in gastric cancer cells or tissues. A schematic representation of factors regulating Cyclin D1 specifically in the context of gastric cancer is provided in Figure 3.

### 4.2. The Regulation of CDK4/6 in Gastric Cancer

CDK4/6 inhibitors have been utilized in various pre-clinical and clinical studies to circumvent the progression of various cancers. However, the treatments lead to resistance, which is one of the major problems that needs to be rectified [107]. The mechanisms of resistance arising against CDK4/6 inhibition in Breast cancer have been reviewed [108]. The status of various inhibitors and clinical outcomes, along with mechanisms of sensitivity and resistance, have also been reviewed in detail [109]. Therefore, an evaluation of literature regarding the regulation of CDK4/6 in gastric cancer is also necessary in order to understand the field of small molecule inhibitors and the ways in which gastric cancer cells may overcome the drug effects using alternative mechanisms of the regulation of these players.

#### 4.2.1. The Regulation of CDK4 in Gastric Cancer

Cyclin-dependent kinase 4 is the first CDK that becomes active as a result of mitogenic signaling upon binding with Cyclin D1. CDK4 inhibition has been at the forefront of the treatments against solid tumors [110]. Gastric inflammatory myofibroblastic tumors can lead to invasion into the diaphragm and spleen, with double amplification of MDM2 and CDK4. The data suggest that CDK4 could be the driver of carcinogenesis in gastric tissues [111]. Pyrotinib is a chemotherapeutic agent explored in clinical trials against gastric cancer. The refraction from Pyrotinib has been associated with the Cyclin D1-CDK4/6 axis; therefore, CDK4/6 inhibitor SHR6390 has also been investigated as a therapeutic option. The combined treatment of the aforementioned agents provided a better response against gastric cancer [92]. P21-activated kinase 1 (PAK1) is involved in gastric cancer progression. The silencing of PAK1 sensitizes cells to CDK4/6 inhibitor in gastric cancer cells in a PDK1-AKT1-dependent pathway [112]. The orphan nuclear receptor (Nurr1) is induced in *H. pylori* infections via PI3K/AKT-Sp1 mediated fashion and in gastric cancer progression, Nurr1 directly binds with CDK4 promoter site for promoting its transcription and facilitating proliferation [113]. *H. pylori* infection in gastric cells leads to the production of Progranulin which autocrine growth factor and it further upregulates CDK4 via PI3K/Akt signaling [114]. A long non-coding RNA GCRL1 promotes the metastasis and proliferation of gastric cancer cells under in vitro and in vivo conditions, where it sponges miR-885-3p to positively regulate the CDK4 levels and its invasion and proliferation-related properties and can be targeted for therapy [115]. CDK4 regulators are summarized in Figure 4.

#### 4.2.2. The Regulation of CDK6 in Gastric Cancer

CDK6 is overexpressed in stomach cancer tissues and is associated with poor survival from the disease. CDK4/6 inhibitor (PD-0332991) targeting CDK6 inhibited development or proliferation and induced apoptosis in stomach cancer cells [116,117]. CDK6 expression has been recorded to be higher in gastric cancer tissues than in normal gastric tissues. Moreover, miR-449a is downregulated in gastric cancer and can target CDK6 in gastric cancer cells directly, as indicated in the gain of function experiments [118]. CDK6 is upregulated in gastric cancer and miR-107 regulates the levels of CDK6 as evidenced by the CDK6 3′UTR luciferase activity in gastric cancer cells [119]. CDK6-specific inhibitor PD0332991 in the gastric cancer cell line led to cell cycle arrest in the G1 phase of the cell cycle with inhibition of pRb at ser780 along with induction of p27 and p53, and therefore has been implicated as a potential therapeutic option [120].

Another study found that Cyclin D1 is upregulated via the Hippo pathway, which is inactivated through LATS2 the promoter of which is hypermethylated due to PAX6 activity. PAX6 activity induces resistance in gastric cancer cells towards Palbociclib targeting CDK4/6. Therefore, it has been suggested to target multiple pathways in order to overcome PAX6-mediated chemoresistance [121]. Cell division cycle 37 like 1 (CDC37L1) is downregulated in gastric cancer and inhibits CDK6, as demonstrated from the experiments showcasing that Palbociclib inversed the effects of CDC37L1 silenced gastric cancer cells [122].

A tumor suppressor miRNA miR1296-5p in gastric cancer cells targets both EGFR and CDK6 for inhibition of growth, invasion, and migration, as observed in MGC-803 and SGC-7901 cell lines [123]. The miR-191-5p targets mRNA of CDK6 coding mRNA at 5′UTR to inhibit the expression of CDK6 and its oncogenic functions in gastric cancer cells [124]. Another microRNA miR-29c that has been shown to be expressed to low extents in gastric cancer tissues and cells targets CDK6 directly, inhibiting its translation and subsequent activity in order to promote the development of gastric cancer [125]. Hyperthermia is one of the adjuvant therapeutic options for gastric cancer treatment [126]. However, gastric cancer cells exhibit resistance to hyperthermia, and CDK6 is upregulated through the induction of AKT pathway inhibition and has been implicated as the protector of cells against apoptosis induced through hyperthermia. Therefore, CDK6 inhibition is considered a relevant option to overcome resistance to hyperthermia-related therapies for advanced gastric cancer [127]. Table 3 displays the molecules studied in gastric cancer that act in regulation of CDK4/6.

A long intergenic non-coding RNA (linc01133) is upregulated in gastric cancer through transcriptional activation of c-Fos and c-Jun. Linc01133 targets miR-145-5p directly for upregulation of YES1 which in turn activates YES-1 dependent YAP1 for upregulating Cyclin D1, CDK4, and CDK6 for promoting the cell cycle transition from G1 to S phase [129]. A circular RNA named hsa-circ-0081143, found in higher concentrations in gastric cancer, sponges miR-646 for its downregulation and induces the expression of CDK6, thus promoting cisplatin resistance in gastric cancer through regulation of the miR-646/CDK6 axis [130,131]. Another circular RNA circZNF609 targeting miR-483 leads to the activation of CDK6 in gastric cancer cells to increase migration and proliferation [132]. Another circular RNA circ_ASAP2 targeting miR-770-5p is upregulated in gastric cancer and leads to activation of CDK6, and has been termed as a potential target for gastric cancer therapy [133]. The UDP-GlcNAc pyrophosphorylase-1 like 1 (UAP1L1) is another upstream driver of CDK6 in gastric cancer, as the phenotypes observed in UAP1L1 overexpressing cells were reversed in CDK6-silenced cells, suggesting that gastric cancer progression depends on CDK6 activity via multiple pathways [128]. Another oncogenic long non-coding RNA LINC00974 upregulates CDK6 in gastric cancer cells and is involved in the regulation of cell cycle progression from G1 to S phase [134]. The factors regulating the CDK6 levels and activity in gastric cancer cells are summarized in Figure 4. All the non-coding RNAs studied in gastric cancer that act as direct or indirect regulators of Cyclin D1-CDK4/6 axis are shown in Table 4.

### 4.3. Cyclin E Regulation in Gastric Cancer

#### 4.3.1. Cyclin E Expression Analyses in Gastric Cancer

Cyclin E1-expressing tumors have been reported to be more invasive compared to those with lower levels of Cyclin E1. Moreover, gastric cancer with aberrant p53 and higher Cyclin E1 can be termed a separate sub-group of the disease, displaying poor prognosis [135]. The lymph node metastases of gastric cancer are correlated significantly with gene amplification in the Cyclin E1 gene. The gastric cancer cell line MKN-7 also exhibits amplification of Cyclin E1 with other cancer lines exhibiting higher protein levels of Cyclin E1. Therefore, it has been suggested that the progression of gastric carcinoma and abnormal growth of gastric cells occur due to events leading to Cyclin E1 overexpression [136]. Cyclin E overexpression and amplification are strongly associated with poor outcomes for gastric cancer patients. HER2 amplification is also correlated with higher Cyclin E expression, and HER2-targeted therapies for gastric cancer lead to Cyclin E-mediated resistance [137].

The Cyclin E1 gene *CCNE1* is expressed highly in gastric cancer tissues, along with gastric cancer cell lines. The higher expression of Cyclin E1 is associated with lymphatic metastases and tumor node metastases, and is associated with poor prognosis of the disease. Moreover, it has been shown that Cyclin E1 inhibition can sensitize cells to cisplatin-induced apoptosis in gastric cancer cell lines such as NCI-N87 and MGC-803 [138]. Cyclin D1 and Cyclin E1 expressions were compared for impact on survival for gastric cancer, and it was observed that Cyclin E-negative cancers portrayed good outcomes, while Cyclin E-positive cancers showed worse outcomes, as indicated through survival curves. Cyclin E overexpression was shown to be a better prognostic factor compared to Cyclin D1 [139,140]. Cyclin E and CDK2 overexpression along with p57 (KIP2) regulation are important factors for metastasis and progression of gastric cancer [141]. Cyclin E-expressing tumors also expressed p53 more than Cyclin E-negative gastric tumors and p53 and Cyclin E-coexpressing tumors were regarded as being more invasive; therefore, Cyclin E is termed an early activator of gastric carcinogenesis [142]. Moreover, the correlation of loss of p21 (*WAF1/CIP1*) and overexpression of Cyclin E has been recorded as a useful prognostic factor for gastric cancer [143]. Another study discovered that lower p27 and higher Cyclin E expression were correlated with poor survival in gastric cancer patients [144].

The low-molecular-weight isoforms of Cyclin E are overexpressed in early-onset gastric cancer (EOGC), and these forms have been associated with survival in EOGC [145]. Cyclin D1 and Cyclin E2 have been studied in the histopathological screening of gastric carcinoma. Both Cyclin D1 and Cyclin E2 were observed to be frequently overexpressed; however, Cyclin D1 was shown to be more sensitive and specific for survival analysis than Cyclin E2 [146]. Moreover, Cyclin E expression was positively correlated with pRB and negatively correlated with p21 in gastric carcinoma cases [147]. A meta-analysis conducted on the literature regarding Cyclin E concluded that Cyclin E overexpression in clinical settings can be used as an indicator of poor prognosis of Gastrointestinal cancer [148]. Since Cyclin E1 expression is high in gastric cancer, its gene amplification (*CCNE1*) was recorded at different metastatic locations, and it was observed that *CCNE1* amplification was significantly associated with liver metastasis of gastric cancer [149]. *H. pylori*-induced precancerosis in gerbils led to a significant increase in Cyclin E1 expression, indicating the role of Cyclin E in the early onset of gastric cancer [150].

As mentioned previously, the CDK4/6 inhibitor Palbociclib is a chemotherapeutic agent utilized to target gastric cancer cells. The sensitivity to Palbociclib depends on Cyclin B1 downregulation as Cyclin E-expressing cells become resistant to CDK4/6 inhibition [151]. Cyclin E-overexpressing cells exhibited high sensitivity to Gemcitabine as compared to Cyclin E-silenced cells, indicating that Cyclin E-expressing cells can be targeted for therapy [152].

#### 4.3.2. Factors Regulating Cyclin E in Gastric Cancer Cells

Cyclin E is regulated through a plethora of upstream factors that can lead to the activation of Cyclin E/CDK2 formation and subsequent progression of cell signaling and the cell cycle [153]. Cell division cycle associated 5 (CDCA5) is involved in the proliferation of gastric cancer cells, and its inhibition results in the downregulation of Cyclin E1 (*CCNE1*). Therefore, it has been suggested that CDCA5 is an upstream activator of Cyclin E1 for the development of gastric cancer [154]. Β-carotene introduction to *H. pylori*-infected cells exhibiting proliferative advantage leads to the downregulation of Cyclin E1, c-myc, β-catenin, and p-GSK3b to inhibit cell proliferation [155]. A long non-coding RNA, GHET1, is upregulated in gastric cancer, and its inhibition in gastric cancer cells leads to reductions in CDK2, Cyclin E1, CDK6, CDK4, and Cyclin D1 [156]. Human ribosomal protein 6 (RPL6) is expressed more highly in gastric cancer and confers multidrug resistance to gastric cancer cells, as observed in cells overexpressing RPL6. Gain-of-function and loss-of-function experiments with RPL6 indicated Cyclin E to be the main target for promoting G1 to S transition for promoting the growth of gastric cancer cells [157].

Since Cyclin E is overexpressed in gastric cancer, its coexpression network analysis provides a unique tool to select the potential oncogenes involved in the progression of gastric cancer. Nuclear transcription factor Y alpha (NF-YA) is found as coexpressed with Cyclin E. Therefore, it was further evaluated in gastric cancer cell lines, and it was observed that NF-YA increases the Cyclin E transcription in gastric cancer to assist gastric cancer progression, and this unique signaling axis can be targeted for therapy as well [158]. An oncogene named 14-3-3ε functions as an upstream factor for Cyclin E activation, as the suppression of 14-3-3ε leads to Cyclin E downregulation in gastric cancer cells [159]. The Enhancer of Zeste Homolog 2 (EZH2), functioning as a polycomb protein, also upregulates Cyclin E in gastric cancer cells, as its inhibition leads to Cyclin D1 and Cyclin E downregulation. EZH2 is found in overexpressed forms in gastric carcinomas and regulates cell cycle-related proteins for the development of invasive phenotypes [160]. Bruceine D (BD) extracted from *Brucea javanica* treatment leads to the inhibition of a long non-coding RNA (LINC01667). LINC01667 is upregulated in gastric cancer, where it sponges miR-138-p to upregulate Cyclin E1 expression. It was observed that BD sensitizes gastric cancer cells to Doxorubicin by targeting LINC01667/miR-138-p/Cyclin E1 axis, and could be a potential drug candidate against gastric cancer [161]. Various factors mentioned here are depicted in Figure 5 to elucidate the findings related to Cyclin E conducted in gastric cancer cells.

### 4.4. Cyclin A and CDK2 Regulation in Gastric Cancer

#### 4.4.1. Cyclin A Expression in Gastric Cancer

Cyclin A expression has been associated with the progression of various cancers. The protein levels of Cyclin A are controlled tightly in order to achieve progression of the cell cycle, and these mechanisms of proteolysis and protein stability demonstrate the involvement of Cyclin A in the progression of normal cells to carcinogenesis [162,163]. In gastric cancer patients, higher expression of Cyclin A has been correlated with HuR (mRNA stability factor). Furthermore, Cyclin A expression and its association with poor survival in gastric cancer are not correlated with distant metastases, but were significantly associated with nodal metastases, penetration depth, noncurative resection, intestinal type, proximal location, high stage, and old age of the patients. The same study concluded that Cyclin A expression was correlated with cytoplasmic localization of HuR and not nuclear HuR, and could be the mechanism for Cyclin A-involvement in the progression or development of gastric cancer [164]. Not many studies have been conducted on the direct involvement of Cyclin A protein involving gastric cancer cells or tissues. However, the Cyclin A partner CDK2 [165] has been studied, and is reviewed in detail.

#### 4.4.2. The Regulation of CDK2 in Gastric Cancer

Generally, Cyclin E and Cyclin A bind to CDK2 sequentially to activate the CDK2, with both Cyclins heading toward ubiquitin-mediated proteolysis. However, CDK2 in unbound form remains inactive and is degraded via a lysosome-autophagy-mediated pathway [166]. As mentioned previously, CDK2 is activated via binding with Cyclin E and Cyclin A for the progression from G1 to S and from S onward in order to maintain cellular integrity [167]. In gastric cancer cells, a long non-coding RNA named LINC01021, which is upregulated in cancer cells, functions in the regulation of KISS1, which is further stabilized via CDK2 activity. CDK2 phosphorylates CDX2 and causes its nuclear export. The mechanism involves LINC01021-mediated binding of CDX2 and CDK2 to promote angiogenesis, invasion, and migration of gastric cancer cells [168]. Moreover, Poly (rC) binding protein 2 (PCBP2), acting as an oncogene in gastric cancer cells, promotes the CDK2 activity by direct interaction and is overexpressed in gastric cancer, where the higher expression is associated with poor survival of the patients [169]. Furthermore, another gastric cancer promoter protein named a DEAD cassette helicase 21 (DDX21), which is an ATP-dependent RNA helicase, is involved in upregulating the expression of Cyclin D1 and CDK2. It provides a rationale for the involvement of CDK2 in the development of gastric cancer as a novel upstream factor for gene regulation of CDK2 [170].

Apart from regular cell cycle regulatory pathways, CDK2 is also a major regulator of cancer cell metabolism in gastric cancer. In gastric cancer cell lines, CDK2 modulates the aerobic glycolytic capacity via suppression of SIRT5 which acts as a tumor suppressor in gastric cancer cells [171]. Another long non-coding RNA, hepatocyte nuclear factor 1 homeobox A antisense RNA 1 (HNF1A-AS1), is upregulated in gastric cancer cells, where early-growth response protein 1 (EFR1) activates the transcription of HNF1A-AS1. HNF1A-AS1 functions as competing-endogenous RNA (ceRNA) to enhance CDC34 expression by binding to miR-661. Subsequently, both EGR1 and HNF1A-AS1 inhibit p21 via activation of CDK2 and CDK4, adding another layer of regulatory mechanisms involved in the function of CDK2 in the development of gastric cancer [172]. Sulforaphane (SFN) exerts antiproliferative effects in gastric effects via inhibition of cell cycle progression and inducing apoptosis and the mechanism includes decreased CDK2 activity p-53-mediated apoptosis to prevent the progression of cells into developing tumorigenic capabilities [173]. Another long non-coding RNA GHET1, which is highly expressed in gastric cancer cells, promotes the cell cycle regulators including CDK2 for inducing the invasive phenotype including the G1-to-S transition and the inhibition of p21 [156].

In gastric cancer cells, TGF-b1 activates caspase-3 to initiate the transition from cell cycle arrest to apoptosis through Rb, p27, and p21 cleavage. These events lead to the activation of CDK2, which has been implicated as a downstream target of caspase to execute the apoptosis induced by TGF-b1 [174]. Furthermore, miR-638 in gastric cancer cells directly regulates the expression of CDK2 [175]. Another micro-RNA, miR-302b, targets CDK2 in gastric cancer cells through ERK signaling [176]. The regulatory factors of Cyclin E-A/CDK2 axis are listed in Table 5, and non-coding RNAs targeting this axis are listed in Table 6. The regulatory factors affecting CDK2 in gastric cancer are displayed in Figure 6.

### 4.5. The Regulation of Cyclin B/CDK1 Axis in Gastric Cancer

#### 4.5.1. Cyclin B1 levels in Gastric Cancer

Cyclin B1, also known as mitotic Cyclin, regulates the progression of the cell cycle at G2/M and is implicated as a driver of tumorigenesis [177]. The lymph node metastasis of gastric cancer has been associated with overexpression of Cyclin B1, with one study showcasing the role of CDK1 and Cyclin B1 pathways in the loss of p27 and progression of gastric cancer [178]. Thirty-two percent of the 61 patients studied exhibited higher levels of Cyclin B1 levels in gastric cancer using immunohistopathological screening [179]. The mRNA of *CCNB1* (Cyclin B1) was also significantly higher in gastric cancer tissues compared to normal tissues, as evidenced by The Cancer Genome Atlas (TCGA) and Oncomine datasets. The same study highlighted the association of higher Cyclin B1 levels with poor overall survival of the disease [180]. The poor prognosis and lymph node metastasis for intestinal-type carcinomas were also linked with the expression of Cyclin B1, as both pRB and Cyclin B1 were positive for diffuse carcinomas [181].

#### 4.5.2. Cyclin B1 Regulation in Gastric Cancer

Cyclin B1 is another Cyclin that functions to controll the cell cycle. It has been implicated in various cancers as a modulator of tumorigenesis and in some resistance to therapy [182]. In gastric cancer cells, Cyclin B1 expression is mechanistically regulated via the interaction of Aurora B with CREPT/RPRD1B. Gastric carcinogenesis is critically regulated through the transcription of Cyclin B1, which is further promoted via Aurora B, which phosphorylates CREPT/RPRD1B. This mode of Cyclin B1 stabilization is known to exhibit the potential to be used as a therapeutic target [183]. The transcription factors involved in Cyclin B1 regulation, determined on the basis of enrichment analysis, are E2F4, NFYA, SIN3A, and FOXM1 [180]. The promoter activity and subcellular distribution of Cyclin B1 are regulated via another oncogene product, Pak1, which is overexpressed in gastric cancer. Pak1 regulates the mRNA and protein levels of Cyclin B1 for the promotion of progression and metastasis of gastric cancer as its knockdown resulted in the inhibition of the growth of gastric cancer cells and xenograft tumors. The mechanism of Cyclin B1 regulation via Pak1 involves the recruitment of NF-kB to the Cyclin B1 promoter site [184]. Cyclin B1 levels are also regulated post-transcriptionally via miR-663, which is a micro-RNA contributing to the suppression of the growth of gastric cancer cells [185]. A transcription factor named Islet-1 (ISL-1), containing a LIM-home domain, is known to function in the progression of gastric cancer. CDK1 phosphorylates ISL1 at S269 in vivo, which further strengthens its binding to Cyclin B1 and Cyclin B2 promoters to accelerate transcription. Moreover, the phosphorylation of ISL-1 via the activity of CDK1 stabilizes ISL-1 in gastric cancer cells [186]. The factors regulating Cyclin B1 in gastric cancer are further presented and summarized in Figure 7.

#### 4.5.3. Regulation of CDK1 in Gastric Cancer

CDK1 is the mitotic CDK that functions to control several important steps of prior to cell division, and has been reported as a target for therapy for various cancers [63,187]. The paracancerous tissues exhibit lower expression of CDK1 compared to gastric cancer tissues, with high CDK1 expressing patients displaying lower survival rates, and CDK1 has been termed an independent prognostic factor for Prostate cancer [188]. Calcium/calmodulin-dependent protein kinase 2 (CAMKK2) works as an upstream factor for CDK1, CDK2, and ERK1 in gastric cancer, as evidenced by bioinformatics analysis. It has been shown that CAMKK2 works in the MEK/ERK1 signaling cascade alongside CDK1 to achieve the progression of gastric cancer [189]. One of the biomarkers of gastric cancer, known as Annexin A4, which functions as an intracellular Ca^2+^ in gastric cancer cells, has been shown to be one of the upstream factors for upregulating the CDK1 mRNA along with other factors such as hyaluronan-mediated motility receptor (RHAMM) and an inhibitor of p21. The major cellular phenotype observed in Annexin A4-expressing cells was the upregulation of epithelial cell proliferation [190].

A long non-coding RNA named CASC11 is also involved in gastric cancer development, with the prime target being miR-340-p. The miR-340-p targets CDK1 directly in gastric cancer cells. Therefore, a mechanism involving the axis of CASC11/miR-340-p/CDK1 has been proposed in which CDK1 functions as an inhibitor of apoptosis and a promoter of the cell cycle [191]. Another transcription factor named Estrogen-related receptor-α (ESRRA) manifests higher in the gastric cancer cell line, and is also known to be an orphan nuclear receptor that targets the DSN1 gene and subsequently enhanced the migration and cell viability of cancer cells via CDC25/Cyclin B1/CDK1 pathway [192]. Cyclin B1-CDK1 complex formation is inhibited as a function of the activity of a tumor suppressor named Ras-associated domain family protein RASSF10, which in gastric cancer is silenced due to hypermethylation of its promoter. RASSF10 functions by promoting GADD45a nuclear accumulation and the induction of mitotic arrest due to the inhibition of the Cyclin B1-CDK1 pathway [193].

Circ_CEA, obtained from Cell Adhesion Molecule 5 (CEA), is another oncogene involved in the progression of gastric cancer with a mechanism involving activation of CDK1 and its subsequent phosphorylation of p53. Circ_CEA functions as a scaffold for enhancing the interaction between CDK1 and p53 and p53^S315^ levels increase as a result of this interaction, which leads to suppression of p53 activity due to reduced nuclear retention and inhibition of the activation of the targets of p53 that can induce apoptosis [194]. In gastric cancer cells, γ-secretase inhibitor (GSI) or Notch inhibition can lead to cell death or mitotic arrest due to the induction of PTEN dephosphorylation at the C-terminus, causing nuclear localization of PTEN. The mechanism involves CDK1, which is one of the substrates of PTEN, where PTEN activation leads to nuclear accumulation of Cyclin B1-CDK1 along with apoptosis [195]. In gastric cancer tissues, CDK1 and CDCA5 expressions are correlated with each other, and this coexpression has been observed in MGC-803 cells, with the inhibition of one or both of these genes leading to suppression of invasion, migration, colony formation, and proliferation [196]. Cyclin B1 is inhibited by the DTW Domain-Containing 1 (DTWD1) tumor suppressor in gastric cancer cells, which is further inhibited via the action of histone deacetylase 3 [197]. The non-coding RNAs targeting the Cyclin B1/CDK1 axis in gastric cancer are listed in Table 7. The upstream factors regulating the Cyclin B1/CDK1 axis are listed in Table 8, and all regulators are shown in Figure 7.

### 4.6. Natural Chemical Compounds against the Cyclin B/CDK1 axis in Gastric Cancer

There are various natural compounds and small molecule inhibitors that have been used to inhibit the Cyclin B1/CDK1 axis to induce mitotic death and cell cycle arrest. Most of these have been reviewed in the context of gastrointestinal cancers [198]. Among natural compounds, flavonoids extracted from *Citrus aurantium* target Cyclin B1 and CDK1 to induce apoptosis in gastric cancer cells, and have been shown to be potential chemoprevention agents [199]. Under both in vitro and in vivo conditions, another specific flavonoid (Kaempferol) leads to Cyclin B1, CDK1, and CDC25C reduction along with increased PARP cleavage and caspase-3 and caspase-9 levels. The same study reported that the inhibitor treatment causes a reduction of COX-2, p-ERK, and p-Akt, implicating the Cyclin B1/CDK1 axis and its involvement in other signaling pathways regulating the growth of gastric cancer cells [200]. Another flavonoid metabolite (2,4,6-Trihydroxybenzoic Acid) has been investigated and observed to be a multiple CDK inhibitor and can be utilized as a chemo-therapeutic agent against gastric cancer [201]. Menadione has been known as an oxidative damage-inducing agent and has been studied in gastric cancer cell lines, where it resulted in proteasome-mediated degradation of Cyclin B1 and CDK1 without reducing the mRNA levels of these genes, with an overall cell cycle arrest in the G2-M phase [202]. Luteolin is another flavonoid that has been investigated for cellular phenotypes in gastric cancer cells, and it also reduced the protein levels of CDC25C, Cyclin B1, and CDK1, and increased the levels of CDK inhibitor p21 with the induction of apoptosis indicated by levels of p53, Caspase 3, 6, 9, and Bax [203].

Cyclin B1 has been shown to be downregulated upon treatment with Ramson watery extract in AGS cells with no impact on G1-related proteins and induction of G2/M arrest and apoptosis [204]. Oridonin is a natural compound extracted from *Rabdosia rubescens* and in gastric cancer cells (SGC-7901); it blocks the cells at the G2/M phase, reducing both Cyclin B1 and CDK1 levels [205]. Cytotoxic licorice compounds have shown promising effects as anti-cancer agents, and Licochalcone (LCA) has been studied for its cytotoxic effects in gastric cancer cell lines. This resulted in apoptosis and decreased expressions of Cyclin B1, MDM2, and Cyclin A1 in MKN-45, AGS, and MKN-28 cell lines [206]. Silbinin extracted from milk thistle is a known flavonolignan compound, and in the gastric cancer cell line (MGC-803), it negatively influenced the STAT3 pathway to activate caspase 3 and caspase 9 activities to induce apoptosis with a negative impact on survival, Cyclin B1, and CDK1 [207]. These data indicate that the cell survival-related properties of Cyclin B1 are also linked with the STAT3 pathway. Resveratrol is another natural compound belonging to the polyphenol family that has been investigated in gastric cancer cells (AGS), leading to the activation of various signaling pathways including PERK/eIF2a and ATF4/CHOP and cell cycle arrest at G2/M and endoplasmic reticulum stress-mediated apoptosis. Moreover, the same study demonstrated that resveratrol inhibited Cyclin B1/CDK1 complex and sensitized cells to cisplatin in a synergistic manner [208].

### 4.7. Other Cyclins and CDKs Investigated in Gastric Cancer Cells

#### 4.7.1. Other Cyclins in Gastric Cancer

Apart from Cyclin D, E, A, and B, as mentioned previously, there are other members of the Cyclin family of proteins that have been studied in the context of gastric cancer. Cisplatin resistance is one of the major hurdles in gastric cancer interventions [209]. Cyclin C is another type of Cyclin that has been studied in the context of drug resistance in gastric cancer. Cisplatin treatment leads to cytoplasmic retention of Cyclin C, which remains anchored at the mitochondria, leading to ROS synthesis and mitochondrial fission. HACE1 adds ubiquitin to Cyclin C for its degradation in gastric cancer cells when treated with cisplatin. Therefore, cisplatin-mediated degradation of Cyclin C is the mechanism of apoptosis in gastric cancer which can be impaired upon mutations in ubiquitinating sites of Cyclin C [210]. Cyclin C regulates cisplatin sensitivity in gastric cancer cells [211].

Cyclin G2 was observed in gastric cancer patients with an exception of female patients, where it was recorded as being lower, and the overall expression was inversely correlated with advanced stages of gastric cancer [212]. Another study explored Cyclin G2 in gastric cancer and observed decreased Cyclin G2 expression in gastric cancer tissues. Overexpression of Cyclin G2 led to a decrease in metastasis and tumor growth under in vitro and in vivo conditions. Cyclin G2 inhibited the β-catenin inhibition with the mechanism depending on the interaction of Dpr1 with Cyclin G2. DPr1 inhibited Wnt/β-catenin signaling via CK1 phosphorylation, which was impacted by Cyclin G2 [213]. GSK-3b is considered to be an upstream factor of Cyclin G2 activity in the aforementioned mechanism of gastric cancer regulation. Cyclin G2 is downregulated in gastric cancer and is regulated via miR-340 which is overexpressed and promotes gastric cancer. In experiments involving the depletion of Cyclin G2, the cellular phenotypes of silenced miR-340 were eradicated whereas CCNG2 3′UTR luciferase activity exhibited direct regulation of Cyclin G2 via miR-340 [214]. Therefore, Cyclin G2 downregulation via miR-340 is one of the mechanisms of how Cyclins are regulated to progress the normal cells to cancerous cells in gastric cancer. Another study investigated the effects of Cyclin G2 overexpression in gastric cancer cells and found that it causes the inhibition of proliferation. Moreover, mutational data were investigated with no major mutations observed; however, there were four synonymous SNPs in Cyclin G2 in gastric cancer samples. It was concluded that Cyclin G2 downregulation in gastric cancer could be the result of other signaling events involved in the development of gastric cancer [215].

Cyclin H is another type of Cyclin that forms a complex with CDK7. CDK7 functions by forming a complex with the RING finger protein known as Mat1 to regulate the cell cycle along with transcription machinery [216,217]. CDK7 inhibition using a specific inhibitor named THZ2 results in the generation of ROS, induction of cell cycle arrest, and apoptosis in gastric cancer cells in in vitro and in vivo conditions. Therefore, CDK7 has been implicated as a therapeutic target for gastric cancer [218]. Cyclin L2 is another Cyclin implicated in cell cycle regulation and transcription of gastric cancer cells. Interestingly, Cyclin L2 overexpression results in cell cycle arrest and inhibition of growth in gastric cancer cells. It also sensitizes gastric cancer cells to cisplatin, docetaxel, and fluorouracil [219]. Cyclin T2 is another type of Cyclin, and has been observed in gastric cancer samples in overexpressed form as compared to normal counterparts. miR-216 is depleted in gastric cancer patients and its overexpression results in the inhibition of invasion, migration, and proliferation along with the induction of apoptosis and cell cycle arrest in gastric cancer cells. Cyclin T2 overexpression reversed the effects of miR-216b mimicking effects in experiments on gastric cancer cells, implicating a mechanism of Cyclin T2 regulation in the development of gastric cancer [220]. Therefore, Cyclin G2 and Cyclin L2, along with Cyclin C, are involved in negatively influencing the growth of gastric cancer cells.

#### 4.7.2. Other CDKs in Gastric Cancer

CDK4/6, 2, and 1 are not the only CDKs regulating the cell cycle. There are other CDKs that depend on various Cyclins to activate their kinase domains [221,222]. CDK5 has been specifically observed to be coexpressed with chromosomal maintenance 1 (CRM1) in gastric cancer, and this coexpression has been described as a promising prognostic model for gastric cancer [223]. In gastric cancer tissues, CDK5 levels are downregulated, and this downregulation is correlated with the severity of the disease as observed in lymph node metastases of gastric cancer. Moreover, the nuclear accumulation or localization of CDK5 is also reduced in gastric cancer cells. When nuclear-targeted CDK5 was ectopically expressed in gastric cancer cells, it led to the inhibition of the proliferation of these cells in a xenograft model. CDK5 inhibition upon treatment with small molecule inhibitor NS-0011, which increases the CDK5 localization in the nucleus, causes suppression of tumorigenesis and proliferation in xenografts [224]. CDK5 interacts with Protein phosphatase 2A (PP2A), which is downregulated in gastric cancer, and for which lower expression is associated with poor survival of the patients. PP2A inhibition in cells stably expressing CDK5 showcased the opposite effects of CDK5-mediated inhibition of gastric cancer progression [225].

Moreover, the CDK7-specific inhibitor BS-181 has also shown promising results against gastric cancer cells, leading to reduced invasion, migration, and proliferation in the gastric cancer cell line [226]. CDK7 expression levels have been correlated with matrix metalloproteinase 14 (MMP14) in gastric cancer, and higher expression of these proteins, along with the mRNA of their genes, has been associated with lymph node matastasis of gastric cancer and worse prognosis of the disease [227]. Moreover, miR-107 also targets CDK8 in gastric cancer cells at its 3′UTR site and regulates the mRNA levels of CDK8 [228]. CDK8 levels and their association with β-catenin delocalization have also been associated with poor prognosis of gastric cancer in a study involving patient tissues and gastric cancer cell lines [229].

CDK9 is another type of CDK that is upregulated in gastric cancer, and is regulated via the activity of miR-613, which functions as a tumor suppressor in gastric cancer cells. Dual luciferase reporter activity demonstrated that miR-613 directly targets the CDK9 gene for its downregulation and suppresses the metastases and progression of gastric cancer cells [230]. CDK10 expression is significantly lower in gastric cancer tissues than in normal tissues. Overexpression of CDK10 led to inhibition of invasion, migration, and proliferation of gastric cancer cells whereas the knockdown of CDK10 demonstrated the opposite phenotypes, indicating that CDK10 is downregulated in gastric cancer. Moreover, loss of CDK10 was also associated with poor survival, tumor differentiation, and metastasis in gastric cancer [231,232]. CDK10 activity and its expression levels in various cancers have been reviewed and it has been shown that this particular CDK behaves as either an oncoprotein or tumor suppressor depending on the tissues; however, in gastric cancer, the levels of CDK10 are lower than in normal tissues [221].

CDK12 expression is high in some cancers and lower in some cancers, and in the case of gastric cancer, it has been observed to be downregulated, and is negatively correlated with poor outcomes, poorly differentiated adenocarcinoma, and advanced stage; therefore CDK12, could be a tumor suppressor in gastric cancer [233]. The CDK12 gene has been reported to be one of the driver genes that regulates the growth of gastric cancer. In gastric cancer cells, CDK12 phosphorylates PAK2 for activation of the MAPK signaling, and its specific inhibitor Proterol has been approved for clinical trials by the FDA due to positive outcomes obtained from PDX and cell line studies [234]. HMGA2 is an oncogene that promotes gastric cancer through a progression of S-G2/M transitions and targets CDK13. HMGA2 and CDK13 are coexpressed in gastric cancer, and higher expression of both these genes has been associated with poor prognosis. SR-4835 is an inhibitor of CDK12/13, and can target HMGA2, thereby indicating a novel mechanism of transition of gastric cancer cells [235]. CDK14, also known as Pftk1, is also overexpressed in gastric cancer, and is involved in promoting metastasis and invasion of tumors [236]. CDK18 is expressed at higher levels in gastric cancer cells, where CXXC finger protein 4 (CXXC4) overexpression can assist in the inhibition of immune escape via the CDK18-ERK1/2 axis. Therefore, CDK18 has been implicated as an immune modulator for tumor progression [237]. Isochorismatase domain-containing 1 (ISOC1) is upregulated in gastric cancer, where it functions to activate and upregulate CDK19 in order to induce proliferation of gastric cancer cells and increase tumor size [238]. The expression levels of all Cyclins and CDKs investigated under tumor vs. normal conditions in gastric cells or tissues are summarized in Table 9.

## 5. Discussion and Future Perspectives

The cell cycle regulation pathways are at the epicenter of cancer biology research for various reasons, including the fact that the major regulators of the cell cycle are prime targets for therapy. The CDKs target multiple pathways involved in the progression of cancer, from benign outgrowths to malignancies. *CDH1* mutations are strongly associated with the development of gastric cancer [253]; however, there is no strong evidence of the involvement of Cyclins or CDKs in *CDH1*-mutated gastric cancers, which can be studied in the future. For example, cyclins and CDKs are mostly upregulated in gastric cancer; however, no correlation has been made between the degree of this upregulation and the driver mutations in the *CDH1* gene [254]. Therefore, correlation studies on mutational analysis for genes of Cyclins or CDKs with *CDH1* can help determine the direct target for therapy especially in advanced cases as there are CDK inhibitors available that are considered potential options as chemotherapeutic agents [76,198].

Different Cyclin-CDK complexes can be sensitized to common factors that tend to function in activating these complexes to transition and progress the cell cycle, such as PI3K/Akt and β-catenin-GSK3β pathways [88,97,155,255]. Moreover, Cyclin E amplification, being associated with gastric cancer progression, provides an opportunity to target multiple CDK inhibitors, such as AZD5438, to target *CCNE1* overexpressing cells more efficiently [256]. Palbociclib has been utilized as a prime inhibitor targeting CDK4/6, inducing senescence in gastric cancer cells [257] and also overcoming 5-FU resistance [151]. Moreover, Aurora B stabilizes Cyclin D1 [91] and also Cyclin B1 [183] through various mechanisms. Aurora B overexpression in gastric cancer cells leads to aneuploidy along with chromosomal instability [258], and also promotes epithelial-to-mesenchymal transition to induce metastasis [259]. Therefore, Aurora B inhibition [260,261] can be utilized as an option for targeting the downstream activation of Cyclin D1 and Cyclin B1, as well, both of which are upregulated in gastric cancer.

Cyclin D1 is the most studied cyclin in gastric cancer cells and tissues. Five micro-RNAs target Cyclin D1 in gastric cells to inhibit the translation and restrict the proliferative capacity of these cells. Since microRNAs present mechanisms that can be explored as therapeutic targets in gastric cancer [262], new common targets can be explored, which can sustain and stabilize these post-transcriptional negative regulators of Cyclin D1, especially in order to overcome the resistance to CDK4/6 resistance. Moreover, three long non-coding RNAs upregulate Cyclin D1, with LINC0857 also positively regulating Cyclin E1 to induce growth of gastric cancer cells (Figure 3). From this data, new joint inhibitory mechanisms for G1 and S phase Cyclins (D and E) can be discovered, as long non-coding RNAs can affect multiple pathways [263,264]. Therefore, in light of the aforementioned implications, and also the use of long non-coding RNAs as potential biomarkers for gastric cancer [265], combinatorial targets can be explored for gastric cancer treatment in future. The epigenetic regulation of Cyclins could be the next step for determining their roles and pathways in gastric cancer progression. In this study, we reviewed 10 microRNAs that directly target the CDK4/6 pathway (Figure 4), implying that CDK4/6 activities are tightly regulated in normal gastric cells and upon oncogenic transformation, the epigenetic control is lifted, leading to activation of CDK4/6 and their target genes. Cyclin D1-CDK4/6 complexes, therefore, present a prominent target for therapy [117], as more studies are being conducted to decrease the knowledge gaps in the context of increasing the efficacy of CDK4/6 inhibition as a primary therapeutic option [13,92,266]. More studies focused on joint targets of miRNAs and non-coding RNAs implicated in altering the Cyclin D1-CDK4/6 axis could lead to an enhanced understanding of the epigenetic mechanisms of gastric cancer development.

Since amplification of the gene giving rise to Cyclin E was associated with gastric cancer, many studies have been carried out to determine the correlation between Cyclin E expression and the prognosis of gastric cancer [135,137,148,246,247]. More importantly, Cyclin E expression also imparts resistance to CDK4/6 inhibition [151]. Therefore, it is recommended to study the combined inactivation of CDK4/6 and CDK2 as an option to overcome this problem. Studies on novel compounds inhibiting multiple kinases can provide better tools as a solution, while upstream factors such as Aurora B, as mentioned before, present better options, as well. Moreover, three long non-coding RNAs upregulate Cyclin E1 in gastric cancer, and these can also be targeted (Figure 5). Cyclin A is the least-studied Cyclin of the four major Cyclins regulating the cell cycle. Since Cyclin A primarily activates CDK2 and helps in S phase progression, its regulators must be studied to decipher the comprehensive cell cycle regulation of gastric cancer. The levels of CDK2 remain unchanged during the development of gastric cancer [221]. Furthermore, there are three long non-coding RNAs and two microRNAs that are implicated as direct regulators of CDK2 in gastric cancer (Figure 6). Apart from these epigenetic regulators, there is a dearth of knowledge regarding the pre-translational regulation of CDK2 in gastric cancer, as only a few modulators of CDK2 were found during the literature search conducted in compiling this study. Therefore, more studies on the regulation of CDK2, especially its activation via Cyclin A, need to be performed to understand the molecular progression of gastric cancer.

The Cyclin B1-CDK1 axis controls the G2-M transition and mitotic exit in gastric cancer regulated via upstream positive regulators, such as E2F4, Aurora B, and Pak1, among others, and negatively regulated via the action of RASSF10, DTWD1, and PTEN, among others (Figure 7). Like other Cyclin-CDK complexes, it is controlled through the action of various micro-RNAs and other non-coding RNAs for the purposes of inhibition and activation respectively. The inhibitors of this vital axis have been reviewed in the context of gastric cancer [198]. Apart from the RNA-mediated epigenetic regulation, Cyclin B1 is also a target of DTWD1, which is a tumor suppressor. DTWD1 is suppressed upon the action of HDAC3, implying that oncogenic activation of Cyclin B1-CDK1 activity is controlled through histone deacetylase activity and chromatin regulation [197]. There are many epigenetic regulators that play important roles in the progression of gastric cancer [267]. Therefore, it is recommended to observe the effects of other activators of epigenetic mechanisms on the Cyclin B1-CDK1 axis. Furthermore, the Cyclin B1-CDK1 complex has been targeted through various natural compounds to arrive at novel therapeutic options against gastric cancer, as described earlier in this study.

Apart from conventional major Cyclins and CDKs, other Cyclins, such as Cyclin C, G2, H, L2, and T2, and other CDKs, such as CDK5, 7, 8, 9, 10, 12, 13, 14, 18, and 19, have been investigated in gastric cancer cells or tissues. Cyclins C, G2, and L2 are downregulated in gastric cancer, whereas Cyclin T2 is upregulated. On the other hand, CDK5, 10 and CDK12 have been observed to be downregulated, while others are more highly expressed in gastric cancer cells or tissues compared to normal gastric cells. The CDK10 expression data from two studies showed variation and can be studied again to develop consensus. The expression levels of all Cyclins and CDKs in gastric cancer versus their normal counterparts have been summarized in Table 1. The tumor suppressor roles of these various Cyclins and CDKs can be explored further to understand gastric cancer progression. Interestingly, Cyclin C has been implicated as the target of Cisplatin to induce apoptosis in gastric cancer cells [210]. However, there is a lack of literature regarding the upstream activators of Cyclin C, which might be the key to developing new understanding and options against gastric cancer.

## 6. Conclusions

The Cyclin family of proteins acts in an ordered and tightly regulated manner for the transition of the cell cycle with other major functions including regulation of proliferation, metabolism, and cellular signaling. Generally, Cyclins activate specific Cyclin-dependent kinases to conduct the cellular signaling processes. CDKs in turn phosphorylate their target proteins through various mechanisms including cell cycle progression, proteasome-mediated degradation, transcription, metabolism, migration, and cell adhesion. There are various Cyclins and CDKs other than conventional Cyclins D, E, A, and B, along with CDK 4/6, CDK2, and CDK1, which mostly act as promoters of cell cycle-regulated pathways and have been implicated in the progression of tumorigenesis. The studies conducted on generic cell lines and other various cancer cell lines have generated functional data for these regulators of the cell cycle. The pathways investigated in other contexts cannot be completely extrapolated to different situations. Therefore, it is important to provide a contextual rationale to form new hypotheses in a specific set of cells, such as gastric cells or gastric cancer cells. In the progression of normal gastric cells to gastric cancer in tissues, these regulators are generally upregulated, with the exception of a few, to promote proliferation, invasion, migration, and cellular metabolism. Most of these actors are found in higher or stabilized forms except a few, to avoid the cell death pathways and induce the proliferative mechanisms. Recently, due to the focus of research shifting towards non-coding RNAs in the regulation of gene expression, various long non-coding RNAs and micro-RNAs have been shown to modulate the activities of various Cyclins and CDKs in the progression of gastric cancer. This review entails the overall comprehensive picture of the status of the upstream regulators and expressions of Cyclins and CDKs in the progression of gastric cancer, where many signaling axes have been implicated as direct therapeutic targets that can potentially lead to positive outcomes for patients in the future. The problem of drug resistance can also be explored by targeting common pathways implicated in multiple Cyclins and CDKs studied in gastric cancer.

## Figures and Tables

**Figure 1 ijms-24-02848-f001:**
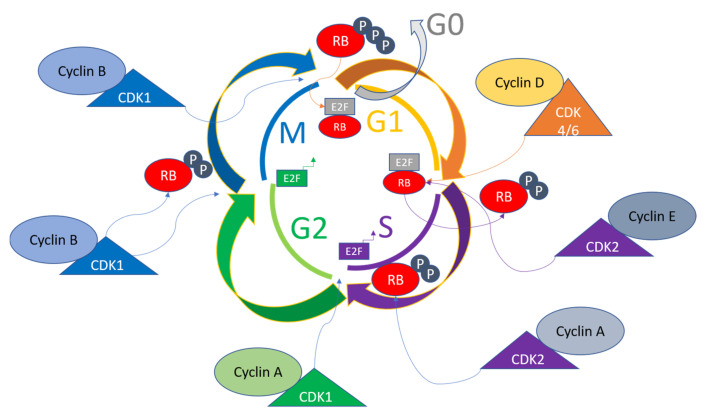
General pathway of cell cycle regulation. Various Cyclins activate different CDKs sequentially to progress the cell cycle and determine the fate of the cells. The CDKs in turn act to phosphorylate various substrates for transitioning the cell cycle phases. For example, RB is one of the major regulators of the cell cycle, and is the target of CDK4/6, CDK2, and CDK1, and together with activated E2F and its target genes, leads to the transition of cell cycle phases.

**Figure 2 ijms-24-02848-f002:**
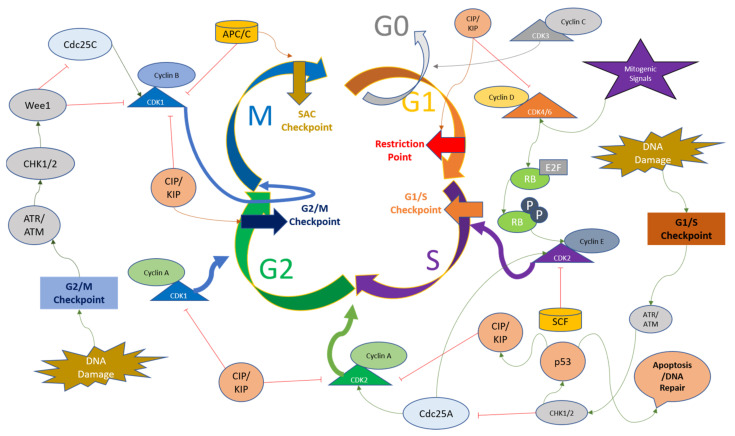
Overview of cell cycle regulation with respect to CDKs and their major regulators. The upstream factors of Cyclins and CDKs can either activate or downregulate their respective functions and cause the cell cycle to halt at various time points including the G1 restriction point, G1/S checkpoint, G2/M checkpoint, and Spindle Assembly Checkpoint (SAC). The description of these factors and their action on Cyclins and CDKs are detailed in the main text.

**Figure 3 ijms-24-02848-f003:**
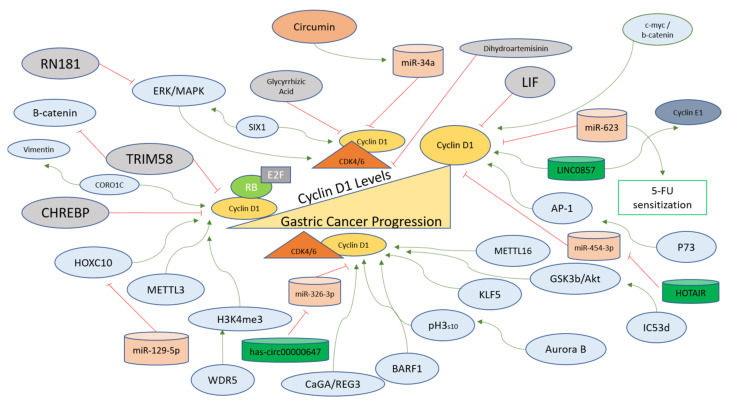
Cyclin D1 regulation in gastric cancer progression. Cyclin D1 levels are upregulated in gastric cancer and are affected positively (green arrows) for the oncogenic transformation of gastric cells via the action of upstream factors (blue). In normal gastric cells, the levels of Cyclin D1 remain tightly controlled via the negative regulators (red lines), which directly inhibit the transcription of the Cyclin D1 gene or inhibit the binding of Cyclin D1 with CDK4/6 directly or indirectly. The modes of action of these regulators are described in the main text.

**Figure 4 ijms-24-02848-f004:**
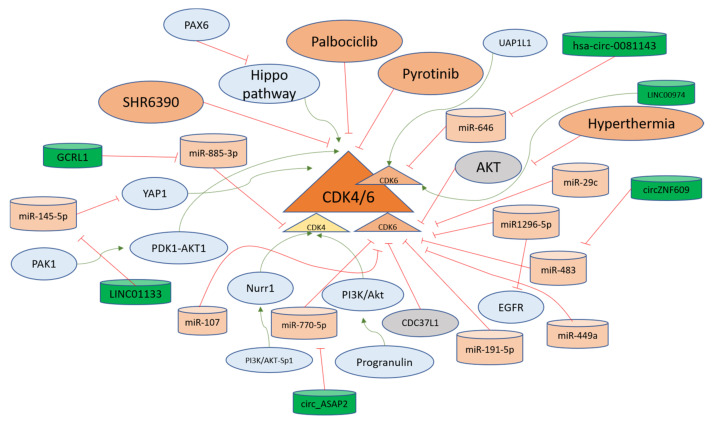
CDK4/6 regulation in gastric cancer progression. CDK4/6 are kinases that accelerate the cell cycle progression from the G1 phase. Different long non-coding RNAs (green) facilitate the upregulation of these CDKs via sponging (red lines) microRNAs (orange). Moreover, other pathways such as PI3K-Akt and Hippo function in activating (green arrows) the CDK4/6 for developing gastric cancer. The negative upstream regulators of CDK4/6, such as CDC37L1 and AKT, are shown in grey and keep the levels of CDK4/6 down in gastric cells. The overall mechanisms for the regulation of CDK4/6 in the gastric cells are explained in the main text.

**Figure 5 ijms-24-02848-f005:**
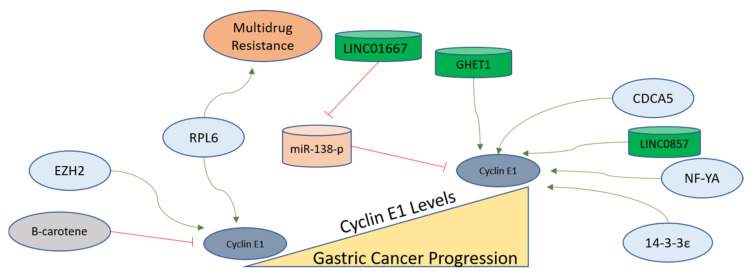
Cyclin E1 regulation in gastric cancer progression. Cyclin E functions to progress cells from G1 to the S phase and remains upregulated in the S phase. In gastric tumors, Cyclin E1 expression is higher than in normal tissues. The factors activating the functions of Cyclin E in gastric cells are shown, with green arrows indicating activation and red lines showing inhibition. The mechanisms of these regulators are described in detail in the main text.

**Figure 6 ijms-24-02848-f006:**
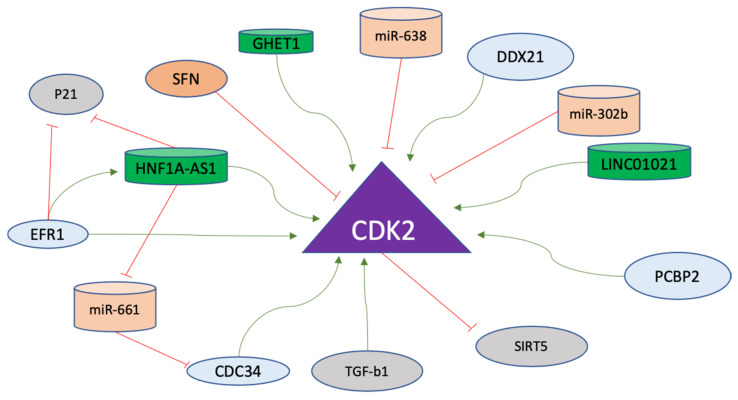
CDK2 regulation in gastric cancer progression. CDK2 is the major driver of the S phase in the cell cycle and promotes various pathways. Positive regulators (green arrows) including long non-coding RNAs (green) and negative regulators of CDK2 (red lines), including micro-RNAs (orange), are shown. The details of the mode of action of these molecules on CDK2 are explained in the text.

**Figure 7 ijms-24-02848-f007:**
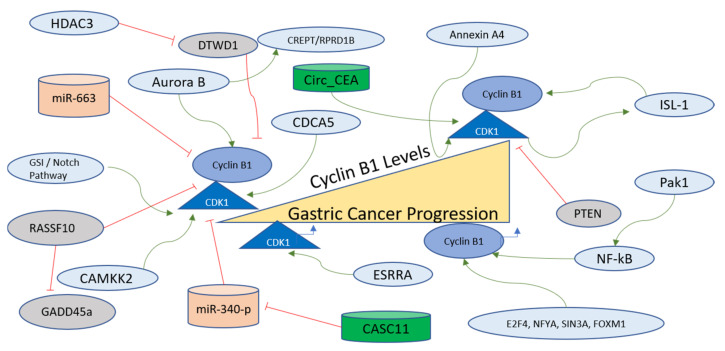
Cyclin B1/CDK1 regulation in gastric cancer progression. The Cyclin B1/CDK1 axis is at the critical conjuncture of G2-M transition and mitotic exit, and is therefore an important component of the cell division pathway. Various factors, including long non-coding RNAs (green), among others (green arrows), activate this axis and are involved in the progression of gastric cancer. The two miRNAs (orange), however, negatively affect the Cyclin B1 and CDK1 in gastric cells, inhibiting their functions and oncogenic properties. Further details of the regulating factors affecting the Cyclin B1/CDK1 axis are described in the main text.

**Table 1 ijms-24-02848-t001:** The upstream molecules or inhibitors downregulating Cyclin D1 in gastric cancer are shown, along with their mode of action.

Regulator Molecule	Cyclin/CDK	Mode of Action	References
CHREBP	Cyclin D1	Negative 242 modulation of the Cyclin D1-Rb-E2F1 signaling mechanism	[82]
RN181	Cyclin D1/CDK4	E3 ubiquitin ligase regulates Cyclin 245 D1-CDK4 activity via inhibition of ERK/MAPK	[83]
Circumin	Cyclin D1/CDK4	Stabilization of miR-34a, which inhibits Cyclin D1 and CDK4	[85]
LIF	Cyclin D1	Downregulates Cyclin D1 and upregulates p21	[87]
TRIM58	Cyclin D1	Reduction of survivin, Cyclin D1, c-myc, and β-catenin via degradation	[88]
Glycyrrhizic acid	Cyclin D1	An anti-tumor compound induces apoptosis along with reduction of Cyclin D1, D2, D3, E1, and E2	[89]
DHA	Cyclin D1/ CDK4	Anti-tumor activities through inhibition of CDK4 activity and targets Cyclin D1 negatively	[90]
AZD1152	Cyclin D1	Specifically inhibits Aurora B also reduces Cyclin D1 for inhibiting tumorigenesis	[91]
SHR6390	Cyclin D1/ CDK4/6	Reduces refraction from Pyrotinib and increases efficacy against gastric cancer	[92]

**Table 2 ijms-24-02848-t002:** The upstream molecules upregulating or activating Cyclin D1 in gastric cancer are shown, along with their mode of action.

Regulator Molecule	Mode of Action on Cyclin D1	Reference
KLF5	Correlated with Cyclin D1 and higher expression showed worse prognosis	[94]
CagA	Assists in formation of Cyclin D1-CDK4 complex formation	[93]
Aurora B	Phosphorylates H3 and induces Cyclin D1 transcription	[91]
METTL16	Upregulates Cyclin D1 transcription through its methyltransferase activity	[95]
BARF1	Stabilizes and interacts with Cyclin D1	[96]
IC53d	Promotes GSK3b/Akt signaling and induces Cyclin D1	[97]
WDR5	Induces H3K4me3 and Cyclin D1 for the progression of the cell cycle	[98]
p73	Regulates activator protein 1 (AP-1) for promoting Cyclin D1	[99]
SIX1	Stabilizes p-ERK and MMP2 for upregulating Cyclin D1	[100]
METTL3	Modifying mRNA via N6-methyladenosine to upregulate Cyclin D1 and activates the Akt signaling pathway	[102]
CORO1C	Assembles F-actin via actin-dependent processes to promote Cyclin D1 and Vimentin	[105]

**Table 3 ijms-24-02848-t003:** The regulators of CDK4/6 investigated in the context of gastric cancer are given. (Note: UAP1L1 is a downstream target of CDK6).

Regulator Molecule	Cyclin/CDK	Effect on Cyclin/CDK	Reference
SHR6390	CDK4/6	Reduces refraction from Pyrotinib and increases efficacy against gastric cancer	[92]
PAK1	CDK4/6	PAK1 silenced cells sensitizes gastric cancer cells ina PDK1-AKT1 dependent pathway to CDK4/6 inhibition	[112]
Nurr1	CDK4	Binds to CDK4 promoter to induce transcription and activation of CDK4	[113]
PD0332991	CDK6	Direct inhibition of CDK6 and reduction of Rb-phosphorylation to induce cell cycle arrest	[120]
PAX6	CDK4/6	Upregulates CDK4/6 and induces chemoresistance in gastric cancer cells against Palbociclib	[121]
CDC37L1	CDK6	Inhibits the activity and expression of CDK6 in gastric cancer cells	[122]
UAP1L1	CDK6	CDK6 regulates UAP1L1-mediated phenotypes (UAP1L1 is a downstream regulator of CDK6)	[128]
PD-0332991	CDK4/6	Direct inhibition of CDK4/6 and induction of apoptosis	[117]

**Table 4 ijms-24-02848-t004:** The non-coding RNAs regulating Cyclin D1-CDK4/6 axis in gastric cancer.

Non-Coding RNA	Category	Cyclin/CDK	Effect on Cyclin/CDK	Reference
miR-129-5p	MicroRNA	Cyclin D1	Downregulation	[84]
miR-34a	MicroRNA	Cyclin D1 CDK4	Downregulation	[85]
miR-623	MicroRNA	Cyclin D1	Downregulation	[86]
LINC0857	Long Non-coding RNA	Cyclin D1	Upregulation	[103]
Hsa_circ00000647	Circular RNA	Cyclin D1	Upregulation	[104]
HOTAIR	Long Non-coding RNA	Cyclin D1	Upregulation	[106]
GCRL1	Long Non-coding RNA	CDK4	Downregulation	[115]
miR-885-3p	MicroRNA	CDK4	Upregulation	[115]
miR-449a	MicroRNA	CDK6	Downregulation	[118]
miR-107	MicroRNA	CDK6	Downregulation	[119]
miR1296-5p	MicroRNA	CDK6	Downregulation	[123]
miR-191-5p	MicroRNA	CDK6	Downregulation	[124]
miR-29c	MicroRNA	CDK6	Downregulation	[125]
LINC01133	Long Non-coding RNA	Cyclin D1-CDK4/6	Upregulation	[129]
circZNF609	Circular RNA	CDK6	Upregulation	[132]
circ_ASAP2	Circular RNA	CDK6	Upregulation	[133]
LINC00974	Long Non-coding RNA	CDK6	Upregulation	[134]
circ-0081143	Circular RNA	CDK6	Upregulation	[131]

**Table 5 ijms-24-02848-t005:** The molecules acting as regulators of Cyclin E-A/CDK2 axis in gastric cancer.

Regulator Molecule	Cyclin/CDK	Effect on Cyclin/CDK	Reference
CDCA5	Cyclin E	Induces proliferative phenotype via Cyclin E activation upregulation	[154]
β-carotene	Cyclin E	Inhibits cell proliferation and downregulates Cyclin E in *H. pylori* infected gastric cells	[155]
RPL6	Cyclin E	Upregulates Cyclin E, confers multi-drug resistance and progresses cells from G1 to S-phase	[157]
NF-YA	Cyclin E	Coexpressed with Cyclin E and increases its transcription	[158]
14-3-3ε	Cyclin E	An upstream factor of Cyclin E, acts as oncogene in gastric cancer	[159]
EZH2	Cyclin E	A polycomb protein upregulates Cyclin E and its inhibition leads to Cyclin D1 and Cyclin E downregulation	[160]
PCBP2	CDK2	Interacts with CDK2 and directly activates it, higher expression is associated with poor prognosis	[169]
DDX21	CDK2	An ATP-dependent RNA helicase directly upregulates Cyclin D1 and CDK2	[170]
EGR1	CDK2	Activates CDK2 and leads to inhibition of p21	[172]
SFN	CDK2	Downregulates CDK2 and induces apoptosis via p53 mediated pathway	[173]
TGF-b1	CDK2	Activates caspase mediated apoptosis and CDK2	[174]

**Table 6 ijms-24-02848-t006:** Non-coding RNAs targeting the Cyclin E/CDK2 axis in gastric cancer.

Non-Coding RNA	Category	Cyclin/CDK	Effect on Cyclin/CDK	Reference
LINC0857	Long Non-coding RNA	Cyclin E1	Upregulation	[103]
GHET1	Long Non-coding RNA	Cyclin E1/CDK2	Upregulation	[156]
LINC01667	Long Non-coding RNA	Cyclin E1	Upregulation	[161]
miR-138-p	MicroRNA	Cyclin E1	Downregulation	[161]
LINC01021	Long Non-coding RNA	CDK2	Upregulation	[168]
HNF1A-AS1	Long Non-coding RNA	CDK2	Upregulation	[172]
miR-638	MicroRNA	CDK2	Downregulation	[175]
miR-302b	MicroRNA	CDK2	Downregulation	[176]

**Table 7 ijms-24-02848-t007:** Non-coding RNAs targeting the Cyclin B1CDK1 axis in gastric cancer are listed.

Non-Coding RNA	Category	Cyclin/CDK	Effect on Cyclin/CDK	Reference
miR-663	MicroRNA	Cyclin B1	Downregulation	[185]
miR-340-p	MicroRNA	CDK1	Downregulation	[191]
CASC11	Long Non-coding RNA	CDK1	Upregulation	[191]
CirC_CEA	Circular RNA	CDK1	Upregulation	[194]

**Table 8 ijms-24-02848-t008:** The upstream regulators of the Cyclin B1/CDK1 axis studied in gastric cancer are listed with their mode of action.

Regulator Molecule	Cyclin/CDK	Effect on Cyclin/CDK	Reference
NFkB	Cyclin B1	Recruited to promoter of Cyclin B1 via Pak1 activity for upregulation of Cyclin B1	[184]
E2F4, NFYA, SIN3A, FOXM1	Cyclin B1	Increase the transcription of Cyclin B1 via transcription factor activity	[180]
Aurora B	Cyclin B1	Interacts with and phosphorylates CREPT/RPRD1B to upregulate transcription of Cyclin B1	[183]
ISL-1	Cyclin B1	CDK1 stabilizes ISL-1, which binds to Cyclin B1 promoter and upregulates its transcription	[186]
CAMKK2	CDK1	Works in MEK/ERK1 signaling cascade and activates CDK1	[189]
Annexin A4	CDK1	Functions as an intracellular Ca^2+^ regulator and upregulates the transcription of CDK1	[190]
ESRRA	Cyclin B1/CDK1	Targets DSN1 and increases cell viability via CDC25/Cyclin B1/CDK1 pathway	[192]
RASSF10	Cyclin B1/CDK1	Promotes the GADD45a nuclear accumulation and inhibits Cyclin B1/CDK1 complex formation	[193]
GSI/PTEN	Cyclin B1/CDK1	GSI dephosphorylates PTEN, which causes nuclear accumulation of Cyclin B1/CDK1 and induces apoptosis	[195]
CDCA5	CDK1	Coexpressed with CDK1 and also stabilizes it in gastric cancer cells	[196]
DTWD1	Cyclin B1	Histone deacetylase 3 inhibits DTWD1 which further inhibits Cyclin B1 in gastric cancer	[197]

**Table 9 ijms-24-02848-t009:** The levels of Cyclins and CDKs alter upon Tumorigenesis. The upregulation or downregulation of all Cyclins and CDKs investigated in gastric cancer tissues or cells compared to their normal counterparts.

Cyclin/CDK	Expression in Cancer vs. Normal Cells/Tissues	References
Cyclin A	Upregulated	[164,239]
Cyclin B1	Upregulated	[178,179,181,240]
Cyclin C	Downregulated	[210]
Cyclin D1	Upregulated	[81,241,242,243]
Cyclin D2	Upregulated	[244,245]
Cyclin E	Upregulated	[135,147,243,246,247]
Cyclin G2	Downregulated	[212,213,214,215]
Cyclin L2	Downregulated	[219]
Cyclin T2	Upregulated	[220]
CDK1	Upregulated	[186,188,196,240,248]
CDK4	Upregulated	[83,249]
CDK5	Downregulated	[224]
CDK6	Upregulated	[250]
CDK7	Upregulated	[218,251]
CDK8	Upregulated	[229]
CDK9	Upregulated	[230]
CDK10	Downregulated	[231,232]
CDK12	Upregulated/Downregulated	[233,234,252]
CDK13	Upregulated	[235]
CDK14	Upregulated	[236]
CDK18	Upregulated	[237]
CDK19	Upregulated	[238]

## Data Availability

Not applicable.

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
