# Peer review of "The Regulation of Cyclins and Cyclin-Dependent Kinases in the Development of Gastric Cancer"

_ijms, 2023, doi:10.3390/ijms24032848_

Round 1
Reviewer 1 Report
In this review paper, the authors have comprehensively reviewed the literature on the regulation of cyclins and cyclin-dependent kinases (CDKs) in the development of gastric cancer. The authors describe the epidemiology, risk factors and classification of gastric cancers, in addition also gave a detailed account of how cyclins and CDKs are up-regulated and or down-regulated at various stages in gastric cancer.
Major comment:
1. Abstract can be condensed as in many places the sentences are repeated in multiple lines.
2. The authors have mentioned the ability of flavonoids to inhibit cyclin B1 and CDK1 (line 665). However, more recent studies show that metabolites of flavonoids (2,4,6-Trihydroxybenzoic acid, 2,4,6-THBA acts as an inhibitor of CDKs (Sankaranarayana et al. 2019; The Flavonoid Metabolite 2,4,6-Trihydroxybenzoic Acid Is a CDK Inhibitor and an Anti-Proliferative Agent: A Potential Role in Cancer Prevention; Cancers 2019, 11(3), 427; https://doi.org/10.3390/cancers11030427). Authors should cite this research paper as it also represents an important study. The review paper will benefit by citation of this research paper.
3. It is not clear why the authors mainly focused on gastric cancer. As far as the regulation of the cyclins and CDKs are concerned, the covered literature is also true for other GI cancers such as colorectal cancers.
Author Response
Author Response Letter
Reviewer 1
In this review paper, the authors have comprehensively reviewed the literature on the regulation of cyclins and cyclin-dependent kinases (CDKs) in the development of gastric cancer. The authors describe the epidemiology, risk factors and classification of gastric cancers, in addition also gave a detailed account of how cyclins and CDKs are up-regulated and or down-regulated at various stages in gastric cancer.
Major comment:
- Abstract can be condensed as in many places the sentences are repeated in multiple lines.
Response: We have condensed the abstract and rectified the redundancy to the best of our ability. The changes are highlighted in the text.
- The authors have mentioned the ability of flavonoids to inhibit cyclin B1 and CDK1 (line 665). However, more recent studies show that metabolites of flavonoids (2,4,6-Trihydroxybenzoic acid, 2,4,6-THBA acts as an inhibitor of CDKs (Sankaranarayana et al. 2019; The Flavonoid Metabolite 2,4,6-Trihydroxybenzoic Acid Is a CDK Inhibitor and an Anti-Proliferative Agent: A Potential Role in Cancer Prevention;Cancers 2019, 11(3),427; https://doi.org/10.3390/cancers11030427). Authors should cite this research paper as it also represents an important study. The review paper will benefit by citation of this research paper.
Response: We have added the article the reviewer suggested in the manuscript. It has increased the quality. The changes are highlighted in the text.
- 3. It is not clear why the authors mainly focused on gastric cancer. As far as the regulation of the cyclins and CDKs are concerned, the covered literature is also true for other GI cancers such as colorectal cancers.
Response: The paper is intended for the Special Issue in the International Journal of Molecular Sciences named ‘Gastric Cancer: Molecular Pathways and Candidate Biomarkers 4.0’. Cyclins and CDKs have rightly been studied as the positive regulators of carcinogenesis in Gastric cancer. Various studies have been conducted to understand the mechanisms of inhibition or activation of these molecules. The upstream effectors have been studied in detail and are searched and summarized in this manuscript with recommendations for future research such as overcoming resistance to inhibitors.

Reviewer 2 Report
In this paper Javed et.al. reviewed the role of regulation of cyclin and cyclin dependent kinases in the development of gastric cancer. This is an exhaustive review with information yet, there is a lot to be improved before acceptance. They are as follows:
More important the paper lacks tables that could streamline and summarize the information discussed in this paper. I would recommend a table for miRNA and create tables for effector proteins. It is so easy to get lost. Also, please write upregulated and downregulated in the table instead of showing arrows. It will be much less confusing to the eyes especially when there is more than one table.
The paper gets more confusing as it carries an explosion of information which makes the reader wonder what is the utility or new about this information. How does it all make sense? This review has collected information from various sources and added into this manuscript but it miserably fails to discuss where does the field stand right now and what is the current status in terms of chemo-therapeutics and overall w.r.t gastric cancer? What are the challenges and how it is being addressed? What are the present therapeutics that are being used that target these pathways? Is it effective? If not, why not? Are there any new drugs targeting these pathways?
Without these information, there are ample reviews that discusses the role of cell cycle in cancer. The USP of this paper will be that it exclusively puts the information in perspective of gastric cancer. This will a highly cited paper if the authors are willing to make these changes.
I look forwarded to the revised manuscript.
Author Response
Reviewer 2
In this paper Javed et.al. reviewed the role of regulation of cyclin and cyclin dependent kinases in the development of gastric cancer. This is an exhaustive review with information yet, there is a lot to be improved before acceptance. They are as follows: More important the paper lacks tables that could streamline and summarize the information discussed in this paper. I would recommend a table for miRNA and create tables for effector proteins.
Response: We have added tables for non-coding RNAs (microRNA, long non-coding RNA) and effector proteins (upstream regulators). The changes and points related to reviewer’s comments are highlighted in the text.
It is so easy to get lost. Also, please write upregulated and downregulated in the table instead of showing arrows. It will be much less confusing to the eyes especially when there is more than one table.
Response: We have added the words (upregulated/downregulated) instead of arrows in the mentioned table. We appreciate the suggestion as it improved the quality of the presentation. The changes are highlighted in the text.
The paper gets more confusing as it carries an explosion of information which makes the reader wonder what is the utility or new about this information. How does it all make sense? This review has collected information from various sources and added into this manuscript but it miserably fails to discuss where does the field stand right now and what is the current status in terms of chemo-therapeutics and overall w.r.t gastric cancer?
Response: Since the aim of this article is to summarize and showcase the relevant research conducted on Cyclins and CDKs in Gastric cancer, the paper is structured from a general to a specific way. Firstly, Gastric cancer-related epidemiology, etiology, diagnosis, and general treatments are provided. Secondly, since Cyclins and CDKs are generally regarded as regulators of the cell cycle, an overview of the cell cycle pathway is given. The main Actors of the paper are both Cyclins and CDKs, therefore their role in the cell cycle is given. Then, the specific section of the paper with multiple subheadings (fifteen) is given. Finally, the content of the aforementioned sections is summarized in a concise manner with specific recommendations for future research.
The sense or utility of the paper comes from the fact that only cyclins/CDKs have been studied a lot in Gastric cancer and this paper is the first of its kind to dive into the subject in a comprehensive manner. Moreover, the structural organization of the paper makes it readable and also understandable, especially for students/researchers who are keen on understanding the general concepts as well. The inclusion of Tables and a later emphasis on therapeutic agents makes it even better. Therefore, we are thankful to the reviewer for their comments and suggestions. The changes and points related to reviewer’s comments are highlighted in the text.
If we look at the data condensed in our paper, we can observe that most of the Cyclins and CDKs are either highly expressed or upregulated in Gastric cancer. Yet, only a handful of axes such as Cyclin D1-CDK4/6 and Cyclin B1-CDK1 have been utilized as targets with small molecule inhibitors. The review does not concern chemotherapeutics. It summarized the literature related to how both Cyclins and CDKS are regulated in Gastric Cancer only. We have already mentioned the CDK inhibitors in the manuscript. Therefore, the logical conclusion from this paper is that the studies have shown consistently that Cyclins/CDKs can be targeted for therapy, but more importantly, the focus of research is to show the factors which regulate these axes as upstream factors with potential as targets for therapy not as direct chemo-therapeutics. The changes and points related to reviewer’s comments are highlighted in the text.
What are the challenges and how it is being addressed? What are the present therapeutics that are being used that target these pathways? Is it effective? If not, why not? Are there any new drugs targeting these pathways? Without this information, there are ample reviews that discusses the role of cell cycle in cancer. The USP of this paper will be that it exclusively puts the information in perspective of gastric cancer. This will a highly cited paper if the authors are willing to make these changes. I look forwarded to the revised manuscript.
Response: Indeed the unique point of our paper is that it dives deep into the literature and stands out as a comprehensive resource for the readers to learn about the studies conducted on the regulation of Cyclins and CDKs solely in the context of Gastric cancer. In fact, the reviewer themselves pointed out the fact that this review can be a highly cited paper. We cannot just include the drugs of the molecules studied in other contexts and include them in our paper as it reduces the uniqueness of the paper as mentioned by the reviewer. We have made the changes the reviewer requested in the revised version. The challenges and therapeutics-related data have been incorporated into the text both in individual sections and also the discussion section as well. The changes and points related to reviewer’s comments are highlighted in the text.
This paper is not about the therapeutic evaluation of Gastric cancer. Instead, the title of the paper is ‘The Regulation Of Cyclins And Cyclin-Dependent-Kinases In The Development Of Gastric cancer’ intended for the Special Issue in the International Journal of Molecular Sciences named ‘Gastric Cancer: Molecular Pathways and Candidate Biomarkers 4.0’. Cyclins and CDKs have rightly been studied as the positive regulators of carcinogenesis in Gastric cancer. Various studies have been conducted to understand the mechanisms of inhibition or activation of these molecules. The upstream effectors have been studied in detail and are searched and summarized in this manuscript with recommendations for future research such as overcoming resistance to inhibitors. This review is also not directly linked to cell cycle regulation and the role of the cell cycle in Gastric cancer. It is about the more commonly studied effectors of Gastric carcinogenesis Cyclins and CDKs both inquiring about the expression status and upstream effectors. The drugs (specific molecule inhibitors) of CDKs investigated in Gastric cancer are both mentioned in individual sections and also in the discussion. The changes and points related to reviewer’s comments are highlighted in the text. We hope our revision and efforts are enough for the reviewer to recommend publication as it certainly has improved the quality of the manuscript.

Round 2
Reviewer 2 Report
The authors have significantly revised the manuscript and have made all the recommended changes. The paper looks much improved compared to the previous version and information is now easy to follow.